# Implication of gut microbiota in the physiology of rats intermittently exposed to cold and hypobaric hypoxia

Sara Ramos-Romero[1,2]*, Garoa Santocildes[2], David Piñol-Piñol[2], Carles Rosés[1], Teresa Pagés[2], Mercè Hereu[1], Susana Amézqueta[3], Joan Ramon Torrella[2], Josep Lluís Torres[1], Ginés Viscor[2]

1 Department of Biological Chemistry, Institute of Advanced Chemistry of Catalonia (IQAC-CSIC), Barcelona, Spain, 2 Physiology Section, Department of Cell Biology, Physiology & Immunology, Faculty of Biology, University of Barcelona, Barcelona, Spain, 3 Departament d'Enginyeria Química i Química Analítica and Institut de Biomedicina (IBUB), Universitat de Barcelona, Barcelona, Spain

* sara.ramos@iqac.csic.es

**Data Availability Statement:** All relevant data are within the manuscript and its Supporting Information files.

## Abstract

This study examines the influence of intermittent exposure to cold, hypobaric hypoxia, and their combination, in gut microbiota and their metabolites *in vivo*, and explores their effects on the physiology of the host. Sprague-Dawley rats were exposed to cold (4°C), hypobaric hypoxia (462 torr), or both simultaneously, 4 h/day for 21 days. Biometrical and hematological parameters were monitored. Gut bacterial subgroups were evaluated by qPCR and short-chain fatty acids were determined by gas chromatography in caecum and feces. Cold increased brown adipose tissue, Clostridiales subpopulation and the concentration of butyric and isovaleric acids in caecum. Hypobaric hypoxia increased hemoglobin, red and white cell counts and Enterobacteriales, and reduced body and adipose tissues weights and Lactobacilliales. Cold plus hypobaric hypoxia counteracted the hypoxia-induced weight loss as well as the increase in white blood cells, while reducing the Bacteroidetes:Firmicutes ratio and normalizing the populations of Enterobacteriales and Lactobacilliales. In conclusion, intermittent cold and hypobaric hypoxia exposures by themselves modified some of the main physiological variables *in vivo*, while their combination kept the rats nearer to their basal status. The reduction of the Bacteroidetes:Firmicutes ratio and balanced populations of Enterobacteriales and Lactobacilliales in the gut may contribute to this effect.

## 1 Introduction

Host health depends largely on the symbiotic benefits contributed by its microbiota. Gut microbiota is a complex microbial community mainly composed of anaerobic bacteria [1]. The host's gut offers an ecological niche to bacterial subgroups, which in turn, provide nutrients and energy to the host through the degradation of dietary polysaccharides. The composition of gut microbiota originally depends on the mode of delivery, maternal microbiota, hygiene of the neonatal environment, use of antibiotic, breast milk or formula regime, and the weaning diet [2]. After the development and diversification of the gut microbiota, its

**Funding:** This work was supported by the Spanish Ministry of Economy, Industry and Competitiveness (DEP2013-48334-C2-1-P, AGL2017-83599-R and graduate fellowship BES2014-068592 to M.H.) and the University of Barcelona (graduate fellowship APIF 2016-2017 to G.S.).

**Competing interests:** The authors have declared that no competing interests exist.

composition depends mainly on the long-term diet of the host [2, 3], and can be also modulated by other environmental factors, such as exposures to cold and hypoxia [4].

Chronic cold exposure modifies the composition of gut microbiota, reducing specifically the Bacteroidetes:Firmicutes ratio and almost eradicating the Verrucomicrobia phyla [5]. It has been hypothesized that gut microbiota shifts induced by cold activate the immune system via: i) production of pro-inflammatory factors as lipopolysaccharides (LPS), and ii) bacterial translocation to the adipose tissue, that could trigger pro-inflammatory ATM1 macrophage infiltration and cytokine release such as TNFα [6]. Changes in gut microbiota induced by cold help to remodel the white adipose and intestinal tissues [5] and upregulate non-shivering thermogenesis [7] in the host. Non-shivering thermogenesis is triggered by prolonged cold exposure, inducing brown adipose tissue (BAT) differentiation [8] and helping to produce heat by catabolizing lipids. Butyrate, a short-chain fatty acid produced by bacterial metabolism [9], regulates feed intake and thermogenesis after cold exposure [7], possibly via the gut-brain axis [1].

The gut microbiota also seems to be related to the host response to hypoxia. Acute mountain sickness is associated to relative abundance of *Prevotella* [10], and it has recently been described that certain patterns of intermittent hypoxia as a therapeutic method can cause alterations in the intestinal microbiota [11–13]. Specifically, 8 h/day of exposure to simulated altitude decreases the density of aerobic bacteria and increases that of *Escherichia coli* (facultative anaerobia) and other anaerobes such as *Bifidobacterium sp.*, *Bacteroides sp.*, *Clostridium perfringens* and *Peptostreptococcus sp.* [11]. The alteration of the intestinal microbiota induced by intermittent hypobaric hypoxia modifies the immune response of the host through at least two ways: i) increasing the production of IgA and IgG antibodies, which helps preventing the entry of microorganisms into the blood stream, and ii) altering the battery of antioxidant enzymes, which increases oxidative stress and activates the immune system [13]. All these pathophysiological changes could be related to disorders associated with acute exposure to hypoxia [11]. In turn, intermittent hypobaric hypoxia has therapeutic applications on muscular injury, bronchial asthma, and several inflammatory pathologies, due to its combined effect at cellular, metabolic and vascular levels [14, 15].

Under natural conditions, cold is an environmental factor associated to hypobaric hypoxia (geographical altitude), so it could be hypothesized that their combination could cause additive or synergistic effects in gut microbiota. Deepen the knowledge on the interaction between cold and hypoxia can contribute to a better understanding of the physiological mechanisms underlying altitude acclimatization. The goal of this study was to explore the effects of intermittent cold and hypobaric hypoxia exposure on gut microbiota of rats, following the application of a protocol that has previously demonstrated efficient results on the recovery of muscle injuries [16].

## 2 Materials and methods

### 2.1 Animals

A total of 52 male Sprague-Dawley rats from a regional supplier of Envigo (Indianapolis, IN, USA), with an initial body weight 215 g (SEM 3), were used. All the procedures strictly adhered to the European Union guidelines for the care and management of laboratory animals and were under license from the Catalan authorities (reference no. 1899), as approved by the University of Barcelona's Ethical Committee for Animal Experimentation.

### 2.2 Experimental design and sample collection

The rats were kept under controlled conditions of humidity (60%), and temperature (22 ± 2˚C) with a 12 h light–12 h dark cycle, following the standard recommendations from European Union commission.

All the groups were fed *ad libitum* with the same standard pellet diet and had free access to water. The animals were randomly divided into four experimental groups (n = 10–14 per group): the control animals (CTL) group; the cold exposed (COLD) group; the hypobaric hypoxia exposed (IHH) group, and the cold plus hypoxia (COHY) group. The hypobaric hypoxia sessions were performed using a hypobaric chamber with a volume of ~450 L, providing space for two cages. A relative vacuum was created by a rotational vacuum pump (TRIVAC D5E; Leybold, Köln, Germany) with its air-flow rate regulated at the inlet by a micrometric valve. Inner pressure was controlled by two differential sensors (ID 2000; Leybold) driving a diaphragm pressure regulator (MR16; Leybold). The target pressure of 462 torr (equivalent to 4,000 m of altitude) was achieved steadily over ~15 min. Once this pressure had been reached, the chamber pressure was maintained and regulated for 4 h/day. At the end of the session, pressurization to normal barometric pressure was gradually restored over 15 min. The cold sessions were performed introducing animals in its cages in a cold chamber (4˚C) for 4 h/day in normoxia or simultaneous to hypobaric hypoxia exposure. The animals were exposed to hypobaric hypoxia and/or cold for 6 days/week, 21 days. Animals had *ad libitum* access to feed and water kept in open-air reservoirs inside the hypobaric chamber during the hypoxia, cold-exposure and normoxia sessions.

Feed consumption and body weight were measured periodically throughout the experiment. At the end of the experiment, rats were fasted overnight and anesthetized intraperitoneally with ketamine and xylazine (80 and 10 mg per kg body weight, respectively). Blood was collected by puncture of the abdominal cava vein and analyzed by a Coulter Spincell 3 (Spinreact; Girona, Spain). Both gastrocnemius muscles, perigonadal adipose tissue (PAT) and interscapular BAT were removed, carefully dissected and weighed. Caecal content and fecal samples were collected and stored at −20˚C until analysis.

## 2.3 Gut microbiota subpopulations

The levels of total bacteria, Bacteroidetes, Firmicutes, Bacteroidales, Clostridiales, Lactobacilliales, Bifidobacteriales, and Enterobacteriales were estimated from caecal and fecal DNA by quantitative real-time PCR (qPCR). DNA was extracted using QIAamp DNA StoolMini Kit from Qiagen (Hilden, Germany) and its concentration was quantified using a Nanodrop 8000 Spectrophotometer (Thermo Scientific; Waltham, MA, USA). All DNA samples were diluted to 20 ng/μL. The qPCR experiments were carried out using a LightCycler 480 II (Roche; Basel, Switzerland) in triplicate. The samples contained DNA (2 μL) and a master mix (18 μL) made of 2XSYBR (10 μL), the corresponding forward and reverse primer (1 μL each), and water (6 μL). All reactions were paralleled by analysis of a nontemplate control (water) and a positive control. The primers, annealing temperatures and positive controls are detailed in S1 Table. The qPCR cycling conditions were: 10 s at 95˚C, then 45 cycles of 5 s at 95˚C, 30 s at primer-specific annealing temperature (S1 Table), and 30 s at 72˚C (extension). Following amplification, to determine the specificity of the qPCR, melting curve analysis was carried out by treatment for 2 s at 95˚C, 15 s at 65˚C, and then continuous increase of temperature up to 95˚C (0.11˚C/s), with five fluorescence recordings per degree Celsius. The relative DNA abundances for the different genes were calculated from the second derivative maximum of their respective amplification curves (Cp, calculated in triplicate) by considering Cp values to be proportional to the dual logarithm of the inverse of the specific DNA concentration, according to the equation: [DNAa]/[DNAb] = 2Cpb-Cpa [17]. Total bacteria were normalized as 16S rRNA gene copies per mg of wet feces (copies per mg).

## 2.4 Short-Chain Fatty Acids in caecal content and feces

SCFAs were analyzed in caecal content and feces by gas chromatography using a previously described method [18] with some modifications. Briefly, the samples were freeze-dried and

weighed (~50 mg dry matter) and a solution (1.5 mL) containing the internal standard 2-ethyl-butiric acid (14.3 mg/L) and oxalic acid (6.0 g/L) in acetonitrile/water 3:7 was added. Then, SCFAs were extracted for 10 min using a rotating mixer. The suspension was centrifuged (5 min, 12,880 g) in a 5810R centrifuge (Eppendorf, Hamburg, Germany) and the supernatant passed through a 0.45 μm nylon filter. An aliquot of the supernatant (0.7 mL) was diluted to 1 mL with acetonitrile/water 3:7. SCFAs were analyzed using a Trace2000 gas chromatograph coupled to a flame ionization detector (ThermoFinnigan, Waltham, MA, USA) equipped with a Innowax 30 m × 530 μm × 1 μm capillary column (Agilent, Santa Clara, CA, USA). Chrom-Card software was used for data processing. This method had already been validated in the working concentration range (acetic and butyric acids 3–750 ppm; propionic acid 1–250 ppm; isobutyric acid 0.3–75 ppm; isovaleric and valeric acids 0.2–40 ppm) [19].

### 2.5 Statistical analysis

All data manipulation, statistical analysis and figures were performed using Graph Pad Prism 5 (Graph Pad Software, Inc., San Diego, CA). The results of the quantitative measurements are expressed as mean values with their standard errors (SEM). Normal distribution and heterogeneity of data were evaluated and statistical significance was determined by one-way ANOVA, and Newman-Keuls' multiple comparisons test was used for mean comparison. Differences were considered significant when $P < 0.05$.

## 3 Results

### 3.1. Feed and energy intakes, and body and organs weights

Feed intake and body weight were monitored throughout the study. Feed intake was similar between the groups (Table 1). Body weight was similar between the groups at the beginning of the study, while the animals in the IHH group significantly reduced their body weight after 21 days of intermittent hypoxia exposure (Fig 1A).

The gastrocnemius muscles weighted significantly less in all the exposed groups than those in the CTL group (Fig 1B). The IHH group significantly ($P < 0.05$) reduced their PAT and BAT weights with respect to the CTL group (Fig 1C & 1D), while the COLD group significantly increased ($P < 0.05$) the weight of their BAT.

**Table 1. Feed intake and hematological values of rats exposed to cold and/or IHH for 21 days.**

|  | CTL | | COLD | | IHH | | COHY | |
|---|---|---|---|---|---|---|---|---|
|  | Mean | SEM | Mean | SEM | Mean | SEM | Mean | SEM |
| Feed Intake (g/day/100 g Body Weight) | 8.5 | 0.5 | 8.8 | 0.5 | 8.0 | 0.4 | 8.4 | 0.3 |
| Red Blood Cells ($x10^3$/μL) | 7.7 | 0.3 | 7.9 | 0.2 | 8.7*[δ] | 0.3 | 9.0**[δδ] | 0.2 |
| Hemoglobin (g/dL) | 13.7 | 0.8 | 15.2 | 0.6 | 17.0* | 0.9 | 16.1* | 0.5 |
| Hematocrit (%) | 48.3 | 2.5 | 51.3 | 1.5 | 56.0 | 2.9 | 56.8* | 2.0 |
| Platelets ($x10^3$/μL) | 502.6 | 44.2 | 590.3 | 41.3 | 498.7 | 32.3 | 447.3[δ] | 26.2 |
| White Blood Cells ($x10^3$/μL) | 5.8 | 0.6 | 6.6[γ] | 0.7 | 9.6* | 1.7 | 6.2[γ] | 0.6 |

Data are presented as means with their standard errors of the mean; n = 10–14 per group. Comparisons were conducted using one-way ANOVA and Newman-Keuls' post-hoc multiple comparisons test.

* $P < 0.05$

** $P < 0.01$ vs CTL group

[δ]$P < 0.05$

[δδ]$P < 0.01$ vs COLD

[γ] $P < 0.05$ vs IHH.

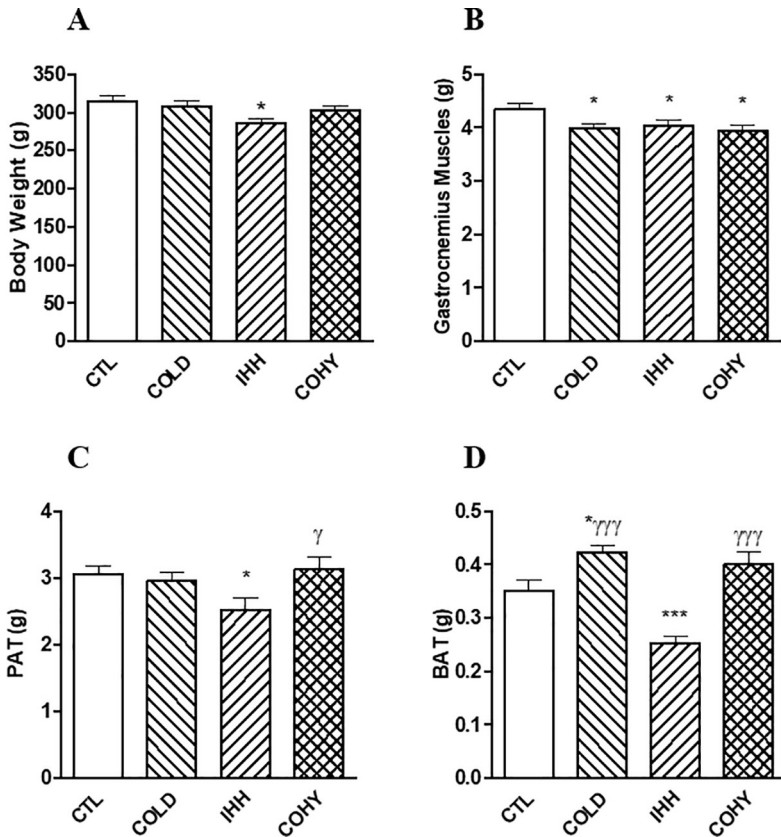

**Fig 1.** Body weight (A), gastrocnemius muscle (B), perigonadal adipose tissue (PAT; C) and interscapular brown adipose tissue (BAT; D) of the experimental groups (CTL, COLD, IHH, COHY) of rats after cold and/or IHH exposure for 21 days. Data are presented as means with their standard error. Comparisons were made using one-way ANOVA followed by Newman-Keuls' post-hoc test. $^*P < 0.05$ and $^{***}P < 0.001$ vs CTL; $^\gamma P < 0.05$ and $^{\gamma\gamma\gamma}P < 0.001$ vs IHH.

## 3.2. Hematology

At the end of the study, the basic hematological values were determined in rats intermittent exposed to 21 days of cold and/or hypobaric hypoxia (Table 1). Animals exposed to hypobaric hypoxia (alone or in combination with cold) presented higher proportion ($P < 0.05$) of red blood cells and hemoglobin than controls. The hematocrit was significantly higher in those animals of the COHY group ($P < 0.05$).

While there was no difference between controls and the other groups in platelet concentration, the IHH group presented higher levels of white blood cells count than the other 3 groups ($p < 0.05$).

## 3.3. Caecal microbiota subpopulations

The relative proportions of the phyla Bacteroidetes and Firmicutes, as well as of the orders Bacteroidales, Clostridiales, Lactobacillales, Bifidobacteriales, and Enterobacteriales, were evaluated at the end of the study (21 days) in caecal content (Fig 2).

The combination of cold and hypobaric hypoxia (COHY group) significantly modified the proportions of the two principal phyla (Bacteroidetes and Firmicutes) of gut microbiota respect to the other experimental groups (Fig 2A and 2B).

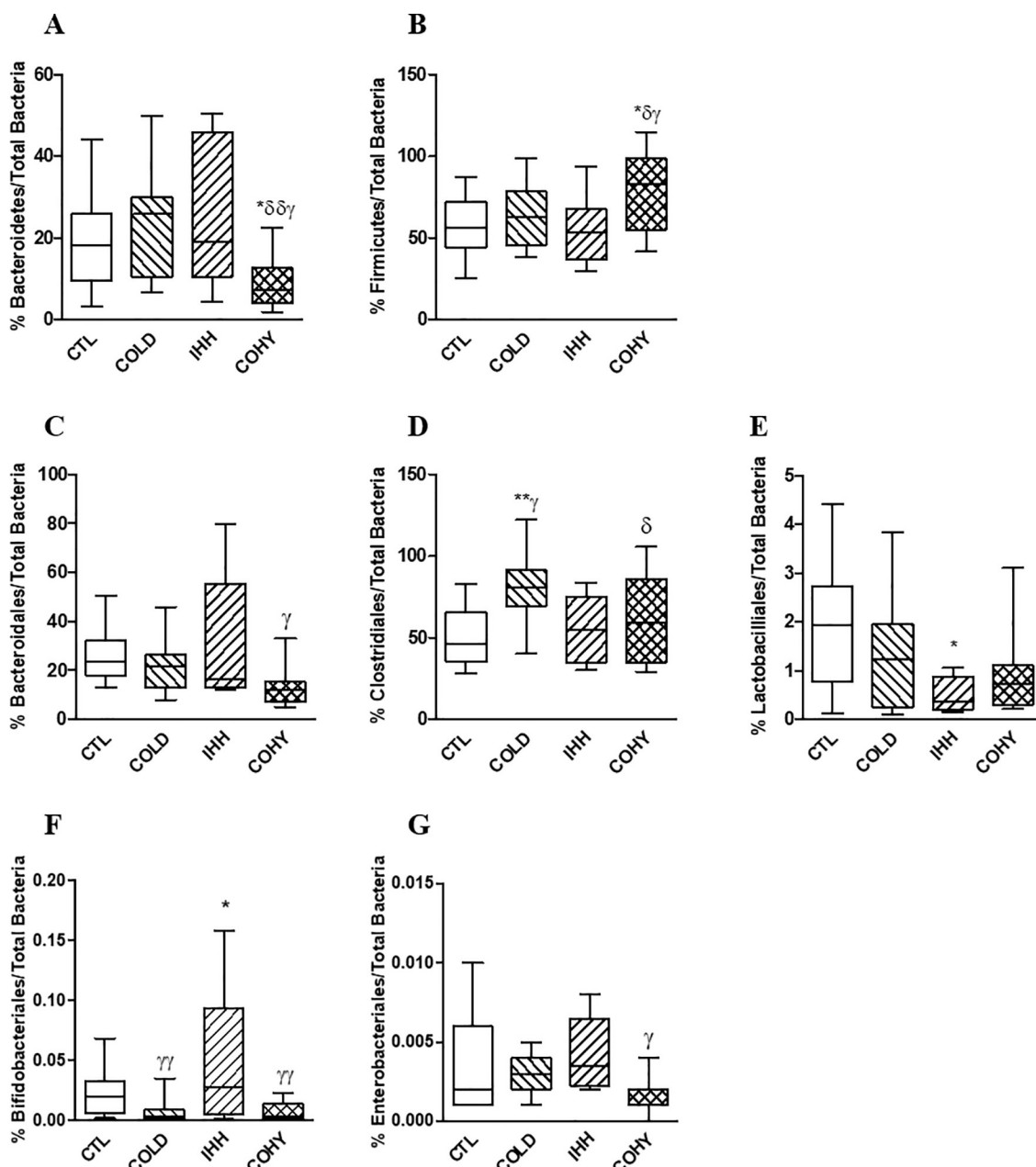

**Fig 2.** Bacteroidetes (A), Firmicutes (B), Bacteroidales (C), Clostridiales (D), Lactobacillales (E), Bifidobacteriales (F), and Enterobacteriales (G) proportions in caecal content samples of the different experimental groups (CTL, COLD, IHH, COHY) after cold and/or IHH exposure for 21 days. Data are presented as box-plots. Comparisons were made using one-way ANOVA followed by Newman-Keuls' post-hoc test. *P < 0.05 and **P < 0.01 vs CTL; $^{\delta}$P < 0.05 and $^{\delta\delta}$P < 0.01 vs COLD; $^{\gamma}$P < 0.05 and $^{\gamma\gamma}$P < 0.01 vs IHH.

Among the Firmicutes phyla, the COLD group increased (P<0.01) the populations of Clostridiales (Fig 2D), while IHH reduced (P<0.05) the proportion of Lactobacilliales (Fig 2E) and increased the proportion of Bifidobacteriales (Fig 2F) with respect to the CTL group. The combined stimuli (COHY group) significantly reduced the Bacteroidales, Bifidobacteriales and Enterobacteriales proportions (Fig 2C, 2F and 2G) compared to the hypobaric hypoxia stimulus alone (IHH group) in caecal content.

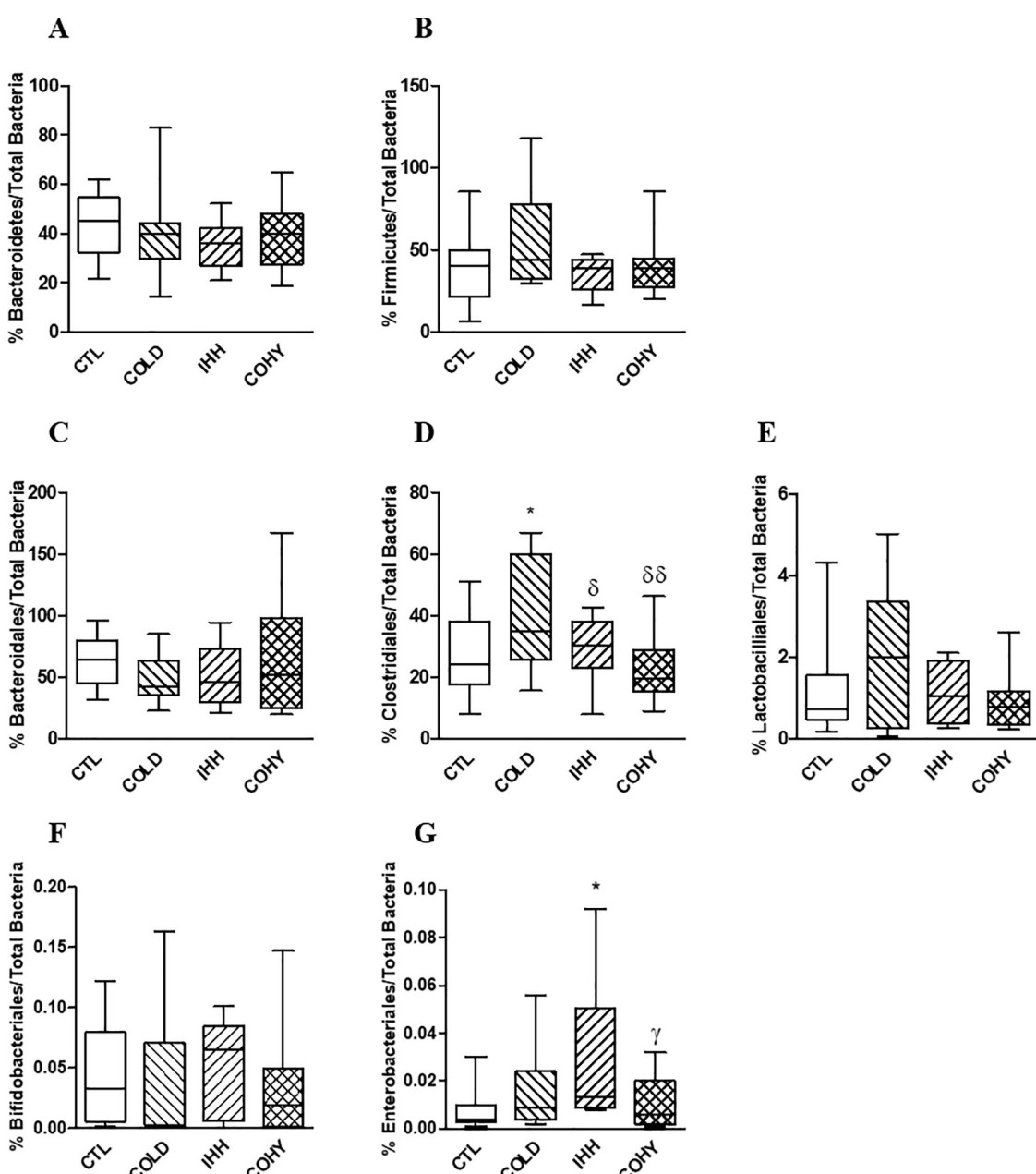

**Fig 3.** Bacteroidetes (A), Firmicutes (B), Bacteroidales (C), Clostridiales (D), Lactobacillales (E), Bifidobacteriales (F), and Enterobacteriales (G) in fecal samples of the different experimental groups (CTL, COLD, IHH, COHY) after cold and/or IHH exposure for 21 days. Data are presented as box-plots. Comparisons were made using one-way ANOVA followed by Newman-Keuls' post-hoc test. $^*P < 0.05$ vs CTL; $^\delta P < 0.05$ and $^{\delta\delta}P < 0.01$ vs COLD; $^\gamma P < 0.05$ vs IHH.

### 3.4. Fecal microbiota subpopulations

The relative proportions of the phyla Bacteroidetes and Firmicutes, as well as of the orders Bacteroideles, Clostridiales, Lactobacillales, Bifidobacteriales, and Enterobacteriales, were evaluated at the end of the study (21 days) in feces (Fig 3).

The cold (COLD group) significantly increased the Clostridiales proportion and the hypobaric hypoxia (IHH group) augmented the Enterobacteriales proportion respect to CTL values

**Table 2. Short-chain fatty acids in caecal content after cold and/or IHH exposure for 21 days.**

| | CTL | | COLD | | IHH | | COHY | |
|---|---|---|---|---|---|---|---|---|
| | Mean | SEM | Mean | SEM | Mean | SEM | Mean | SEM |
| Acetic acid | 214.7 | 16.4 | 263.1 | 22.3 | 209.2 | 14.6 | 253.8 | 27.3 |
| Propionic acid | 35.8 | 4.9 | 27.7 | 2.7 | 25.3 | 2.7 | 31.7 | 4.5 |
| Isobutyric acid | 3.2 | 0.4 | 2.2 | 0.2 | 2.6 | 0.3 | 2.9 | 0.5 |
| Butyric acid | 105.0 | 15.6 | 143.1[γ] | 9.5 | 86.7 | 10.0 | 122.1 | 11.4 |
| Isovaleric acid | 3.3 | 0.4 | 1.9* | 0.2 | 2.7 | 0.4 | 2.5 | 0.4 |
| Valeric acid | 4.3 | 0.5 | 4.3 | 0.3 | 3.5 | 0.5 | 4.6 | 0.6 |
| Total SCFA | 366.4 | 33.5 | 442.3 | 29.6 | 330.1 | 22.5 | 429.5 | 41.8 |

Data are presented as means with their standard errors of the mean; n = 10–14 per group. Short-chain fatty acids (SCFAs) are given as millimoles per kilogram of feces. Comparisons were conducted using one-way ANOVA and Newman-Keuls' post-hoc multiple comparisons test.

\* P < 0.05 vs CTL group

γ P < 0.05 vs IHH group

in feces. The combined stimuli (COHY group) reverted these changes (Fig 3D and 3G). There were no changes between groups in the other studied bacterial subpopulations.

## 3.5. Short-chain fatty acids in caecal content and feces

The concentrations of 6 SCFAs were measured in caecal content and feces at the end (21 days) of the study (Tables 2 and 3).

Cold significantly (P < 0.05) reduced the concentration of isovaleric acid with respect to the control group and butyric acid with respect to the IHH group in caecal content (Table 2). There were no differences in any SCFA among the groups in the fecal samples (Table 3).

## 4 Discussion

The gut microbiota co-develops with the host, and its bacterial proportions are modified by his physiology and the actions of different diets and extrinsic stressors [4]. The present study examines the effects of an intermittent exposure to cold, hypobaric hypoxia or the combination of both stimuli on gut microbiota of healthy rats. As far as the thermo-neutral zone ranges from 27 to 30°C in rats [20], our results on cold exposure are the additive effect of chronic exposition to 22°C (as recommended by EU guidelines for the accommodation and care of

**Table 3. Short-chain fatty acids in feces after cold and/or IHH exposure for 21 days.**

| | CTL | | COLD | | IHH | | COHY | |
|---|---|---|---|---|---|---|---|---|
| | Mean | SEM | Mean | SEM | Mean | SEM | Mean | SEM |
| Acetic acid | 97.1 | 12.5 | 92.9 | 8.6 | 88.0 | 9.3 | 90.7 | 8.5 |
| Propionic acid | 17.6 | 2.5 | 12.3 | 2.5 | 10.9 | 2.6 | 14.3 | 2.1 |
| Isobutyric acid | 1.3 | 0.2 | 1.2 | 0.2 | 0.8 | 0.1 | 1.1 | 0.2 |
| Butyric acid | 28.9 | 5.4 | 41.7 | 8.6 | 21.5 | 5.0 | 25.0 | 2.9 |
| Isovaleric acid | 1.3 | 0.3 | 1.3 | 0.2 | 0.9 | 0.1 | 1.1 | 0.2 |
| Valeric acid | 1.6 | 0.2 | 1.7 | 0.3 | 0.9 | 0.2 | 1.4 | 0.2 |
| Total SCFAs | 147.9 | 20.2 | 143.9 | 17.3 | 123.6 | 14.4 | 132.4 | 13.2 |

Data are presented as means with their standard errors of the mean; n = 10–14 per group. Short-chain fatty acids (SCFAs) are given as millimoles per kilogram of feces. Comparisons were conducted using one-way ANOVA and Newman-Keuls' post-hoc multiple comparisons test.

rodents used for experimental and other scientific purposes) plus intermittent 4°C exposure. It is widely known that cold increases BAT deposition by β-adrenergic activation that leads to increased intracellular cyclic AMP [8]. BAT catabolizes triglycerides to produce heat through non-shivering thermogenesis by uncoupling protein-1 (UCP1, or thermogenin) [8]. It has been recently demonstrated that lack of gut microbiota impairs UCP1-dependent thermogenesis in cold-exposed mice, and that butyrate partially counteracts this effect [7], probably via the gut-brain axis [1]. Most butyrate producers in the gut microbiota belong to the Clostridiales order, in particular to clostridial clusters IV and XIVa [21]. Our results are in accordance with the described role of gut microbiota in thermogenesis, as animals exposed to cold presented more BAT than controls (Fig 1D), as well a higher proportion of Clostridiales in the gut (Figs 2D and 3D) and more butyric acid in caecum (Table 2).

The other condition assayed in the present study was hypobaric (low barometric pressure) hypoxia. Rats intermittently exposed to hypoxia over a period of 21 days presented higher proportions of red blood cells, hemoglobin and hematocrit than controls (Table 1), as a consequence of an increased erythropoiesis, an intermittent hypobaric hypoxia effect widely known [14]. Also, the IHH group presented lower body weight and white adipose tissue than the other groups (Fig 1A and 1C), although no significantly changes in their feed intake (Table 1) were detected. Reduction of body weight as a response to chronic hypoxia exposure has been previously described in humans [22–24]. The main mechanisms behind this hypoxia-induced weight loss at high altitudes (severe hypoxia) are: i) reduction of nutritional energy intake (anorexia), ii) reduction of intestinal energy uptake from impaired intestinal function, and/or iii) increased energy expenditure [22]. Interestingly, in this study hypobaric hypoxia combined with cold (COHY group) counteracted the reduction of body weight and PAT, which were similar to control animals not exposed to hypobaric hypoxia. This effect could be related, at least in part, to the changes in the proportions of Bacteroidetes and Firmicutes in the gut of the COHY group (Fig 2A and 2B). A reduction in the Bacteroidetes:Firmicutes ratio has been linked to an increased capacity of fat storage in both rats and humans [25, 26]. The lipid accumulation associated with these changes in gut microbiota are related to low levels of fasting-induced adipose factor (Fiaf) and phosphorylated AMP-activated protein kinase (AMPK) [27].

Intermittent hypobaric hypoxia also increased the proportions of Enterobacteriales in feces (Fig 3G). In the gut, differentiated colonocytes exposed to low oxygen consumption conditions, such as hypoxia, paradoxically increase the amounts of $O_2$ and $NO_3^-$ in the intestinal lumen [28]. This change in the intestinal environment shifts the gut microbiota community from obligate anaerobes to facultative anaerobes [28] such as Enterobacteriales. Increased proportions of gut Enterobacteriales have been related to low-grade inflammation in the host, at both intestinal and systemic levels [4, 29]. IHH group significantly reduced the amounts of Lactobacilliales with respect to controls (Fig 2E). The reduction of Lactobacilli in the gut of both animals and humans exposed to different types of intermittent hypoxia (episodic, intervallic and chronic) has been widely described [4, 13, 30–33]. Intermittent hypoxia can lead most tissues to increase anaerobic metabolism even in the context of a normal circulatory system at rest. Lactate is the general endpoint of the anaerobic use of glucose in the tissues, consequently, hypoxia exposure and basal increased lactic production could affect the lactic acid microbiota. Lower proportions of Lactobacilli have been also related to intestinal inflammation [4]. Here, the higher proportions of white blood cells corroborated the pro-inflammatory state of the IHH animals (Table 1). Interestingly, this imbalance on gut microbiota populations was avoided when hypobaric hypoxia was combined with cold (COHY group). Cold exposure exacerbates the adrenergic response and the metabolic activation elicited by hypoxia, also causing increased hyperventilation and tachycardia, thus mitigating the drop in the oxygen partial pressure induced by hypoxia [28]. The combination of cold with IHH probably keeps

the intestinal environment closer to normal conditions, avoiding the imbalance of Enterobacteriales and Lactobacilliales that induces a pro-inflammatory state in the host.

Knowledge of the changes induced by cold and hypoxia on gut microbiota may open new avenues for preventing or mitigating the alterations derived from exposure to harsh conditions, by designing specialized diets or food supplementation (if needed) with probiotics, prebiotics or other ingredients with eubiotic action (promotion of a balanced microbiota).

In conclusion, the combination of intermittent cold and hypobaric hypoxia exposure counteracts the hypoxia-induced weight loss, may be related to the reduction of the Bacteroidetes: Firmicutes ratio, and counteracts the hypoxia-induced pro-inflammatory effect, normalizing the populations of Enterobacteriales and Lactobacilliales.

## Supporting information

**S1 Table.**
(DOCX)

## Author Contributions

**Conceptualization:** Sara Ramos-Romero, Joan Ramon Torrella, Josep Lluís Torres, Ginés Viscor.

**Data curation:** Sara Ramos-Romero, Garoa Santocildes, David Piñol-Piñol, Carles Rosés, Mercè Hereu, Joan Ramon Torrella.

**Formal analysis:** Sara Ramos-Romero, Mercè Hereu, Susana Amézqueta, Joan Ramon Torrella, Josep Lluís Torres, Ginés Viscor.

**Funding acquisition:** Josep Lluís Torres, Ginés Viscor.

**Investigation:** Sara Ramos-Romero, Garoa Santocildes, David Piñol-Piñol, Carles Rosés, Teresa Pagés, Mercè Hereu, Susana Amézqueta.

**Methodology:** Susana Amézqueta, Joan Ramon Torrella.

**Project administration:** Josep Lluís Torres, Ginés Viscor.

**Supervision:** Sara Ramos-Romero, Teresa Pagés, Joan Ramon Torrella, Josep Lluís Torres, Ginés Viscor.

**Writing – original draft:** Sara Ramos-Romero.

**Writing – review & editing:** Sara Ramos-Romero, Josep Lluís Torres, Ginés Viscor.

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
