## [Decision Letter · Decision Letter 0]

17 Jul 2020

PONE-D-20-11941

Implication of gut microbiota in the physiology of rats intermittently exposed to cold and hypobaric hypoxia

PLOS ONE

Dear Dr. Ramos-Romero,

Thank you for submitting your manuscript to PLOS ONE. After careful consideration, we feel that it has merit but does not fully meet PLOS ONE’s publication criteria as it currently stands. Therefore, we invite you to submit a revised version of the manuscript that addresses the points raised during the review process.

We look forward to receiving your revised manuscript.

Kind regards,

Juan J Loor

Academic Editor

PLOS ONE

Journal Requirements:

Reviewers' comments:

Reviewer's Responses to Questions

**Comments to the Author**

1. Is the manuscript technically sound, and do the data support the conclusions?

Reviewer #1: Yes

2. Has the statistical analysis been performed appropriately and rigorously? 

Reviewer #1: Yes

3. Have the authors made all data underlying the findings in their manuscript fully available?

Reviewer #1: Yes

4. Is the manuscript presented in an intelligible fashion and written in standard English?

Reviewer #1: Yes

5. Review Comments to the Author

Reviewer #1: Ms “Implication of gut microbiota in the physiology of rats intermittently exposed to cold and hypobaric hypoxia” ，examined the influence of intermittent exposure to cold, hypobaric hypoxia, and their combination to gut microbiota in SD rats. The main findings are that: cold increased brown adipose tissue, Clostridiales subpopulation and the concentration of butyric and isovaleric acids in caecum. Hypobaric hypoxia increased hemoglobin, red and white cell counts and Enterobacteriales, and reduced body and adipose tissues weights and Lactobacilliales. Cold plus hypobaric hypoxia counteracted the hypoxia-induced weight loss as well as the increase in white blood cells, while reducing the Bacteroidetes:Firmicutes ratio and normalizing the populations of Enterobacteriales and Lactobacilliales.

Data is interesting, data analysis and presentation is fine, conclusion is clear. The main problem is that authors only reported the phenotypic changes and lack the mechanism investigation.

I have some comments and suggestions:

1. Rats raised in 22 C temperature seems too low, rats in this ambient temperature facing cold exposure and thus can increase the thermogenesis (such as from brown adipose tissue and other organs) to keep stable body temperature, and/or increase the food consumption. For 4 C exposure, this is the further (or additive effect) low temperature effect.

2.For brown adipose tissue, from which location and how many BAT removed from the body? if showed some data for UCP1 in BAT and non-shivering thermogenesis will be benefit for the conclusion.

3. It is better to do some transplanting experiments, such as transplanting gut microbiota of cold rats to 22C rats, or from hypoxia to normal condition, or the combination.

6. PLOS authors have the option to publish the peer review history of their article (what does this mean?). If published, this will include your full peer review and any attached files.

Reviewer #1: No

---

## [Author Response · Author response to Decision Letter 0]

27 Jul 2020

Dear Editor and Reviewer,

We would like to thank you for your time and effort concerning our manuscript and the constructive criticism we have obtained. We have addressed the concerns raised by the reviewer in our point-by-point answers and done revisions to the manuscript accordingly as indicated in the following answers. 

1. Rats raised in 22 C temperature seems too low, rats in this ambient temperature facing cold exposure and thus can increase the thermogenesis (such as from brown adipose tissue and other organs) to keep stable body temperature, and/or increase the food consumption. For 4 C exposure, this is the further (or additive effect) low temperature effect.

Rats were raised at a standard room temperature of 22 ºC, following the guidelines from European Union and the regulations of the Governments of Catalonia and Spain. Concretely, EU commission recommend rodents should be maintained within a temperature range of 20 ºC to 24 ºC, as follow: https://eur-lex.europa.eu/legal-content/EN/TXT/?uri=celex:32007H0526

The rat’s thermo-neutral zone ranges from 27 to 30 ºC (Gordon, Lee, Chen et al 1991), nevertheless the results obtained in the studied groups are referred to the control group, raised at 22 ºC. In this sense, the results obtained from 4 ºC exposure are an additive effect, as the reviewer pointed out. This is now included in the Discussion section (Lines 243-246)

2. For brown adipose tissue, from which location and how many BAT removed from the body? if showed some data for UCP1 in BAT and non-shivering thermogenesis will be benefit for the conclusion.

The brown adipose tissue collected was the whole interscapular BAT from all animals (now indicated in Line 121). 

We agree with the reviewer about the interest about UCP1 in BAT and non-shivering thermogenesis in this kind of studies. Unfortunately, we have not the techniques available to analyse that in the present experiment. We will include them in future works.

3. It is better to do some transplanting experiments, such as transplanting gut microbiota of cold rats to 22C rats, or from hypoxia to normal condition, or the combination.

Despite the interest of a transplantation experiment, it is not possible to do it without the pertinent approval from the Ethical Committee for Animal Experimentation. For the present project, we cannot ask for such authorization since faecal transplantation was not included in the aims of the funded project.

---

## [Editor Report · Decision Letter 1]

1 Oct 2020

Implication of gut microbiota in the physiology of rats intermittently exposed to cold and hypobaric hypoxia

PONE-D-20-11941R1

Dear Dr. Ramos-Romero,

We’re pleased to inform you that your manuscript has been judged scientifically suitable for publication and will be formally accepted for publication once it meets all outstanding technical requirements.

Kind regards,

François Blachier, PhD

Academic Editor

PLOS ONE

---

## [Editor Report · Acceptance letter]

19 Oct 2020

PONE-D-20-11941R1 

Implication of gut microbiota in the physiology of rats intermittently exposed to cold and hypobaric hypoxia 

Dear Dr. Ramos-Romero:

I'm pleased to inform you that your manuscript has been deemed suitable for publication in PLOS ONE. Congratulations! Your manuscript is now with our production department. 

Kind regards, 

on behalf of

Dr. François Blachier 

Academic Editor

PLOS ONE